# Perceived Help-Seeking Difficulty, Barriers, Delay, and Burden in Carers of People with Suspected Dementia

**DOI:** 10.3390/ijerph18062956

**Published:** 2021-03-13

**Authors:** Carmen K.M. Ng, Dara K.Y. Leung, Xinxin Cai, Gloria H.Y. Wong

**Affiliations:** 1Department of Social Work and Social Administration, The University of Hong Kong, Hong Kong 999077, China; carnghku@gmail.com (C.K.M.N.); daralky@hku.hk (D.K.Y.L.); caixx@hku.hk (X.C.); 2Sau Po Centre on Ageing, The University of Hong Kong, Hong Kong 999077, China

**Keywords:** service accessibility, dementia knowledge, affordability, carer role strain, self-criticism, negative emotions

## Abstract

Because of an often complicated and difficult-to-access care system, help-seeking for people with suspected dementia can be stressful. Difficulty in help-seeking may contribute to carer burden, in addition to other known stressors in dementia care. This study examined the relationship between perceived help-seeking difficulty and carer burden, and the barriers contributing to perceived difficulty. We interviewed 110 carers accessing a community-based dementia assessment service for suspected dementia of a family member for their perceived difficulty, delays, and barriers in help-seeking, and carers burden in terms of role strain, self-criticism, and negative emotions. Linear regression models showed that perceived help-seeking difficulty is associated with carer self-criticism, while carer role strain and negative emotions are associated with symptom severity of the person with dementia but not help-seeking difficulty. Inadequate knowledge about symptoms, service accessibility, and affordability together explained more than half of the variance in perceived help-seeking difficulty (Nagelkerke R^2^ = 0.56). Public awareness about symptoms, support in navigating service, and financial support may reduce perceived difficulty in help-seeking, which in turn may reduce carer self-criticism during the early course of illness.

## 1. Introduction

Dementia is a public health priority that affects more than 50 million people and counting [1,2]. As societies struggle to meet the needs of dementia care [3], it is becoming clear that early intervention and carer support are key [4,5].

Despite increasing attention paid to the positive gains associated with caregiving [6], dementia carer burden and stress are widely studied and documented [7,8]. Based on the stress process model [9], dementia carer stress has been conceptualized as comprising: background and context of care (e.g., service accessibility), primary stressors (e.g., behavioral symptoms and functioning of the person with dementia), secondary strains (e.g., carer intrapsychic strains), mediators or moderators of stress (e.g., coping and social support), and outcomes or manifestations of stress (e.g., carer health and nursing home placement of the person with dementia) [7]. The conceptualization has facilitated significant development in understanding dementia carer needs and support service development.

Previous studies, however, focused mainly on carers of people with diagnosed dementia. Carers are often the first to notice symptoms of dementia and actively involved in help-seeking [10,11,12,13,14]. Help-seeking is often defined as the active process of seeking help when the person with dementia and the carer became aware of the symptoms of dementia [15]. The caring process therefore may begin way before a diagnosis is confirmed: there can be considerable time lapse before help is reached, ranging from 2 to 4 years [9,13,14,16,17]. In a study examining the clinical profile and symptom recognition of help-seekers in Hong Kong, where a community-based early detection program for dementia has been established, the median time to seek help was 1 year with moderate cognitive impairment upon presentation [18], while an earlier Hong Kong study in memory clinics reported a median duration of 2 years before seeking medical consultation and at an advanced stage [19]. Although there is evidence suggesting a longer delay in help-seeking is associated with worse functioning and symptoms [18], it is yet unknown whether this delay and difficulty in help-seeking is related to greater carer burden, although reduced quality of life has been observed in carers very early even when the illness is mild [20].

Emerging evidence suggests that the pre-diagnostic and help-seeking period in dementia can be a long and difficult experience [21]. In case of denial and resistance of the person with suspected dementia to seek help, carers may be trapped in the moral dilemma of respecting their autonomy or intervening by seeking professional help [22,23]. On the other hand, help-seeking may also be complicated by provider- or system-level barriers, with difficult-to-access service and long waiting time being commonly reported problems in dementia care [24]. Other barriers in help-seeking include insufficient knowledge about dementia, stigma, and fear [25,26]. Previous reviews also identified the following barriers: misconception of dementia as normal ageing, beliefs in non-medical causes, regarding dementia as personal or family responsibility, previous experience of shame, sense of helplessness, adverse experience with healthcare service, and inability to recognize symptoms of dementia [15,22]. Although there is currently no universally agreed way to categorize these barriers, a recent review using thematic analysis method suggested the following categories of barriers: (1) denial, (2) stigma and fear, (3) lack of knowledge, (4) normalization of symptoms, (5) preserving autonomy (related to resistance in seeking help), (6) lack of perceived need, (7) unaware of changes, (8) lack of informal network support, (9) carer difficulties, and (10) problems accessing help [27].

This study aims to explore the relationship between carer burden and delay, perceived difficulty, and barriers in help-seeking in carers seeking help from a community-based early detection program for dementia, among other key stressors in caregiving, to suggest further developments in early intervention and carer support. The specific objectives are to (1) examine the contribution of help-seeking factors in carer burden on top of the primary stressors of behavioral symptoms and functioning; and (2) identify the more salient reported barriers contributing to help-seeking difficulty.

## 2. Materials and Methods

This is a cross-sectional study using interview and service data collected from an early detection program for dementia between 2015 and 2018 in Hong Kong. The program has been described elsewhere [18,28]; briefly, it is a community-based, territory-wide, open-referral service for people with suspected dementia or their carers, who can make an appointment to arrange for pre-diagnostic assessment by trained professionals in dementia care. The service was set up to improve accessibility of specialist diagnostic service by shifting part of the assessment burden to community-based social care setting. In this study, trained researchers independent from the service conducted face-to-face semi-structured interviews with carers.

### 2.1. Participants

Participants were dyads with a family carer aged 18 years or above and identified him/herself as the primary carer of the person with suspected dementia. A joint informed consent from the dyad was required to participate in the study. Only those whose assessment results subsequently led to a confirmed diagnosis of dementia were included. A total of 110 dyads of persons with dementia and their family carers were included.

### 2.2. Measures

#### 2.2.1. Time to Help-Seeking 

Time to help-seeking (in months) was calculated by comparing time of recognition of the earliest symptom and date of first help sought from a formal service [18,28]. Time of recognition of the earliest symptom was determined by first asking family carers an open-ended question “What triggered you to seek help?”. Spontaneous complaints were then categorized into memory decline, impaired judgment, impaired understanding, getting lost in familiar places, not recognizing friends or family, impaired verbal ability, hallucination, delusion, and others. Carers may report more than one compliant. For each compliant, the interviewer then asked about the month and year when it was first noticed. To determine the date of first help sought from a formal service, carers were then asked whether they had sought help from a list of formal and informal help providers, which included public outpatient clinic, private outpatient clinic, emergency room, other professionals in private practice (e.g., occupational therapists), statutory or non-governmental organization, immediate family members, other relatives and friends, and others. For each type of help sought, the interviewer then asked about the month and year when they first got in contact for the identified symptom. Help sought from a public or private doctor, emergency room service, other professionals in private practice, and statutory or non-governmental organization were categorized as formal service.

#### 2.2.2. Perceived Difficulty and Barriers in Help-Seeking 

Carers were asked to rate their subjective feeling of difficulty in the help-seeking process on a 5-point Likert scale from very easy to very difficult. They were then asked to report any perceived barriers to their help-seeking with the following options with reference to known categories, with some regrouping for eliciting the underlying concepts from the help-seeking carer perspective, taking into account some categories cannot be elicited by direct questioning (e.g., denial): lack of time to seek help (reflecting carer difficulty), concern over fees (reflecting carer difficulty), concern over stigma (stigma and fear), disproval by family or friends (reflecting preserving autonomy), lack of knowledge about symptoms (reflecting lack of knowledge, normalization of symptoms, lack of perceived needs, unaware of changes, beliefs in non-medical causes), difficulty in accessing service (reflecting problems accessing help), having to rely on oneself (reflecting lack of informal network support, sense of helplessness), unwillingness to disclose the person’s problem (reflecting stigma and fear), and others (open-ended category to capture barriers not previously reported). Carers may report more than one barrier.

#### 2.2.3. Carer Burden

We used the 12-item Cantonese short version of the Zarit Burden Interview (CZBI-Short) to assess carer burden. The CZBI-Short was validated locally and shown to consists of three factors, namely role strain, feelings of self-criticism, and negative emotions [29]. In this sample, the Cronbach’s alpha for the three subscales was 0.88, 0.94, and 0.77, respectively.

#### 2.2.4. Other Stressors

We used the Alzheimer’s Disease Cooperative Study-Activities of Daily Living (ADCS-ADL) [30] to assess self-care ability and daily functions of the person with dementia. It is an informant-based interview with 23 items covering the physical and mental functioning, and independence in self-care of the person with dementia. Behavioral and psychological symptoms of dementia (BPSDs) were assessed using the Hong Kong Chinese version of the Neuropsychiatric Inventory-Questionnaire (NPI-Q) [31], a 12-item informant-based interview on common BPSDs. The NPI-Q has a severity subscale and a carer distress subscale. In the current analysis, only the severity subscale was used, to reflect symptom severity, and to minimize the potential conceptual overlap between carer distress due to BPSDs and carer burden.

### 2.3. Data Analysis 

To explore sample characteristics associated with perceived difficulty and time to help-seeking, we conducted one-way analysis of variance (ANOVA) by levels of perceived help-seeking difficulty on global burden and each of the three carer burden subscales (role strain, feelings of self-criticism, and negative emotions), and bivariate correlations were computed. Linear regression models were used to test the effects of perceived help-seeking difficulty on carer burden subscales. We then conducted an ordinal regression analysis to investigate the respective contribution of individual barriers on perceived help-seeking difficulty. Complete case analysis (listwise deletion for missing data) was used in all analyses.

## 3. Results

### 3.1. Sample Characteristics

Table 1 shows the characteristics of the sample. Most carers in this study were men in their late 50 s, who were adult children of the person with dementia in their late 70 s, mainly women. The median time to help-seeking was 12 months, with a wide range between 0 and 204 months. In this sample of help-seekers accessing an early detection service, majority perceived the help-seeking process to be average in difficulty. Considering the response distribution, two levels of perceived help-seeking difficulty “difficult” and “very difficult” were combined in subsequent data analysis. Among the perceived barriers, lack of knowledge about the symptoms was the most frequently reported, found in nearly half of the population.

### 3.2. Exploratory Analyses on Perceived Difficulty and Time to Help-Seeking

One-way ANOVA suggested that among participants who reported different level of perceived help-seeking difficulty, there were differences in their level of role strain (df = 3, F = 8.02; *p* < 0.001), self-criticism (df = 3, F = 8.53; *p* < 0.001), and negative emotions (df = 3, F = 6.44; *p* < 0.001), and the symptom severity of the person with dementia (df = 3, F = 3.85, *p* = 0.012). Specifically, carer scores for role strain were higher for those who rated the help-seeking process “difficult or very difficult” (9.29 ± 6.48) than those who rated the process “easy” (3.07 ± 3.79) or “fair” (3.34 ± 3.99). Negative emotions were higher for those who rated “difficult or very difficult” (5.65 ± 3.10) than those who rated “easy” (2.11 ± 2.10) or “fair” (3.18 ± 2.69). Self-criticism were higher for those who rated “difficult” or “very difficult” (4.29 ± 2.20) than for those who rated “very easy” (1.11 ± 1.57), “easy” (1.54 ± 1.93), or “fair” (2.20 ± 2.25).

Bivariate correlations (see Appendix A for correlation matrix of key variables) showed that a longer time to help-seeking is associated with more severe BPSDs (*r* = 0.36, *p* < 0.01), while greater perceived help-seeking difficulty is associated with worse ADL (*r* = 0.25, *p* < 0.05), more severe BPSD (*r* = 0.20, *p* < 0.05), greater self-criticism (*r* = 0.41, *p* < 0.01), and more negative emotions (*r* = 0.23, *p* < 0.05).

### 3.3. Linear Regression Models of Carer Role Strain, Self-criticism, and Negative Emotions

Three linear regression models were fitted to identify predictors including help-seeking factors for carer role strain, self-criticism, and negative emotions (Table 2, Table 3 and Table 4). Characteristics of the persons with dementia, carer characteristics, and time and perceived difficulty in help-seeking were included as predictors. In the carer self-criticism model, the predictors accounted for 36% of the variance (Table 2). Carer perceived difficulty of help-seeking was the only significant predictor in the model. In the model for role strain, the predictors explained 52% of the variance (Table 3). In the model for negative emotions, the predictors explained 46% of the variance (Table 4). The person’s BPSD severity was the only significant factor contributing to carer role strain and negative emotions. In all three models, variance inflation factor values for predictors included ranged between 1.07 and 1.26, suggesting no serious issue with multicollinearity.

### 3.4. Ordinal Regression Model of Perceived Help-Seeking Difficulty

Table 5 shows the ordinal regression model to identify predictors including reported barriers for perceived help-seeking difficulty. When characteristics of the person with dementia, carer, and perceived barriers were considered in the model, the factors together explained over half of the variance (pseudo R^2^ = 0.56; *p* < 0.001). Three perceived barriers were significant in the model, namely cost concerns, inadequate knowledge about symptoms, difficulty in accessing service.

## 4. Discussion

To our knowledge, this is the first study exploring the role of help-seeking factors in carer burden before a dementia diagnosis. Our findings provided support to earlier observations that carer burden occurs in the pre-diagnostic period, and suggested perceived help-seeking difficulty as a related stressor on top of other known stressors such as BPSDs and ADL functioning of the person with dementia, as shown in the regression model for self-criticism, where perceived difficulty in help-seeking significantly contributed to carer self-criticism, while BPSDs contributed to role strain and negative emotions but not self-criticism. The results further illustrated the contributions of barriers (symptom knowledge, affordability, and service access) in the perceived difficulty in help-seeking.

Our finding that perceived difficulty of help-seeking is specifically associated with carer self-criticism is worth noting for future research on intrapsychic strains. Although BPSDs, a well-established primary stressor in dementia carer burden, were associated with carer role strain and negative emotions, they did not contribute to self-criticism in our sample. A possible explanation is that carers regard help-seeking for the person with dementia as their responsibility, and difficulties encountered in the process may be internalized, even when the difficulties are caused by external factors (e.g., service access) beyond their control. On the other hand, carers may not consider BPSDs to be their responsibility in the same way they do with help-seeking. Without proper support and guidance, caregiving can be an experience of self-failure, frustrations, and doubts of one’s competence [9], and our findings suggest this may particularly be the case for tasks perceived as a carer responsibility. Difficulties encountered in the help-seeking process in general, largely related to an inaccessible service system, may contribute to higher carer burden way before diagnosis and support can be provided.

The finding that perceived help-seeking difficulty contributed to carer burden in terms of self-criticism deserves attention to the possible nature of their relationship. In a previous study, carers have described the pre-diagnostic period to be tense, and full of conflict, anger and misunderstanding among family members [21]. These tensions could be due to the power struggle among family members over the decision to seek professional help, especially when the person with dementia or other family members were in denial [11,32], or conflict between the person with dementia and their adolescent grandchildren who did not understand that some behaviors were due to their illness. In a recent qualitative study seeking to understand carer experience of transitioning into their carer roles, the pre-diagnostic period was described as full of negative emotions and self-doubt. Recalling the difficult times in the help-seeking process evoked strong emotions in some of the carer participants who became tearful in a focus group discussion [33]. A key feature of the stress process is stress proliferation. People are seldom exposed to one single stressor as stressors tend to precipitate and lead to other stressors. Secondary strains, of which self-criticism is a part of it, are the cognitive, affective and physiological distress induced by the primary stressors [34]. Once established, they exist as independent stressors and can be as influential as the primary stressors in creating stressful outcomes [7]. When applied to understanding dementia carer burden, the early intrapsychic strains of self-criticism arising in the help-seeking process could perpetuate other stressors, which future research on early intervention designed to target help-seeking experience should investigate.

Although inadequate knowledge about the symptoms was the most common barrier reported by almost half of the carers, a perceived inaccessible service was the most significant barrier associated with carer perceived difficulty in help-seeking. It is known that service systems are overly complex and difficult to access from the carer perspective [25,35]; our findings showed that this is a key reason for a perceived difficulty in help-seeking, highlighting a need to address this problem in efforts to promote early intervention in dementia. Public education needs to provide not only information about symptom recognition, but also direct and simple message about where help is available.

Although not shown to be directly related to the help-seeking process itself, carer burden in terms of role strain and negative emotions can be observed in this group of help-seeking carers, which can be explained by BPSDs that occur before diagnosis. This is not surprising, given that BPSDs have been found to be the strongest predictor of carer burden [36,37] and nursing home placement of the person with dementia [38]. The service implication nevertheless is still linked with earlier help-seeking and intervention: without appropriate carer support during the pre-diagnostic period, carers may lack the skills and knowledge in handling BPSDs, potentially increasing the distress associated with the BPSDs. On the other hand, a longer wait before seeking help may also be a consequence of more severe BPSDs, especially in those with frontotemporal dementia, which may not be recognized as dementia symptoms and interpreted as maladaptive interpersonal problems [39], again highlighting the barrier of lack of symptom knowledge in help-seeking.

This study has several limitations. First, because of feasibility consideration, a convenient sample from an early detection service was used; a lack of random sample limits the extent to which the results can be generalized to other context and populations. The sample was also not sufficiently powered to test a larger set of predictors in the regression models, or to employ more advanced statistical methods to investigate the potentially complex inter-relationship between variables. The findings therefore need replications in future studies. In particular, findings from this early detection sample may differ from those accessing traditional specialist diagnostic care, and the ease of service access may be higher by design. Second, as the interviews happened only after the dyads have reached the service, findings may be affected by recall bias as the carers were asked to report when they first noticed symptoms and to comment on the level of difficulty of the help-seeking process retrospectively. Third, the cross-sectional nature of the current exploration precluded conclusions regarding the direction of any relationships observed. The findings of an association between perceived difficulty of help-seeking and carer burden may reflect the effects of the latter on subjective perception; in other words, a reverse causation cannot be ruled out. Although this does not fully explain the specific association between perceived help-seeking difficulty and carer self-criticisms but not role strains and negative emotions, the findings should be treated as preliminary. Finally, the current study focuses on an early detection service that formed only part of a triage system, before a person reaches clinical diagnosis stage, the impact of a definitive diagnosis has on carer burden and perceived help-seeking difficult remains unknown from our findings. Further studies are needed to explore the impact of the diagnostic process. These findings nevertheless provided basis for designing and conducting longitudinal studies, with larger samples sufficiently powered to allow simultaneously testing the various interacting variables based on the stress process model, using more advanced statistical methods such as structural equation modelling, and incorporating help-seeking factors identified in this study.

## 5. Conclusions

This study provided initial evidence of the roles of help-seeking experience in contributing to carer burden, on top of the known primary stressors associated with dementia such as BPSD. Our findings suggest that public awareness about symptoms, support in navigating service, and financial support may reduce carer burden, which is linked with perceived difficulty in help-seeking. The potential impact of these strategies as part of the wider early intervention initiatives for dementia should be further investigated.

## Figures and Tables

**Table 1 ijerph-18-02956-t001:** Sample characteristics.

	Person with Dementia	Carer
Age, years, mean (SD)	78.7 (9.0)	57.5 (14.3)
Female gender, *n* (%)	75 (68.2)	41 (37.3)
Education level, *n* (%)		
No formal education ^1^	51 (47.2)	3 (3.0)
Primary	31 (28.7)	17 (16.8)
Secondary	20 (18.5)	46 (45.5)
Post-secondary or above	6 (5.6)	35 (34.7)
Carer relationship, *n* (%)		
Adult children	69 (62.7)	
Spouse	34 (30.9)	
Grandchildren	3 (2.7)	
Others (other relatives, friends)	4 (3.6)	
ADCS-ADL, mean (SD) (range, 1–76)	50 (18.9)	-
NPI-Q severity, mean (SD) (range, 0–26)	4.18 (5.78)	-
CZBI-short, mean (SD) (range, 0–48)	-	10.9 (9.3)
Role strain (range, 0–24)	-	4.7 (5.1)
Self-criticism (range, 0–8)	-	2.13 (2.25)
Negative emotions (range, 0–12)	-	3.42 (2.83)
Perceived help-seeking difficulty, *n* (%) ^2^		
Very difficult	-	2 (1.9)
Difficult	-	15 (14.3)
Average	-	41 (39.0)
Easy	-	28 (26.7)
Very easy	-	19 (18.1)
Perceived barriers to help-seeking, *n* (%)		
Lack of knowledge about the symptoms		41 (48.8)
Difficulty in accessing service		13 (15.5)
Lack of time to seek help		9 (10.7)
Relying on oneself		8 (9.5)
Cost concerns		7 (8.3)
Disproval by family or friends		3 (3.6)
Unwillingness to disclose the person’s problem		2 (2.4)
Concern over stigma		1 (1.2)
Time to help-seeking, months, median (range)	12.0 (0–204)	

^1^ Data missing in two persons with dementia (*n* = 108) and nine carers (*n* = 101) for this item. ^2^ Data missing in five participants for this item (*n* = 105). ADCS-ADL = Alzheimer’s Disease Cooperative Study-Activities of Daily Living; NPI-Q = Neuropsychiatric Inventory-Questionnaire; CZBI-short = Cantonese short version of the Zarit Burden Interview.

**Table 2 ijerph-18-02956-t002:** Linear regression models of carer self-criticism.

Predictors	*β*	*SE*	*t*	*p*
Person with Dementia Characteristics				
Age	0.02	0.03	0.48	0.63
Gender (male)	0.41	0.70	0.59	0.56
ADCS-ADL	0.01	0.02	0.78	0.44
NPI-Q severity	0.03	0.05	0.53	0.60
Carer Characteristics				
Age	−0.02	0.03	−0.85	0.40
Gender (male)	−0.07	0.62	−0.11	0.92
Education	−0.01	0.29	−0.04	0.97
Monthly income	0.00	0.00	0.40	0.69
Help-seeking Factors				
Time to help-seeking	0.01	0.01	0.46	0.65
Perceived difficulty	1.17	0.30	4.0	<0.00

*R*^2^ = 0.36, F(10, 42) = 2.35, *p* = 0.026. ADCS-ADL = Alzheimer’s Disease Cooperative Study-Activities of Daily Living; NPI-Q = Neuropsychiatric Inventory-Questionnaire.

**Table 3 ijerph-18-02956-t003:** Linear regression models of carer role strain.

Predictors	*β*	*SE*	*T*	*p*
Person with Dementia Characteristics				
Age	0.00	0.06	0.03	0.98
Gender (male)	−0.19	1.37	−0.14	0.89
ADCS-ADL	−0.01	0.03	−0.44	0.66
NPI-Q severity	0.41	0.10	4.05	<0.00
Carer Characteristics				
Age	−0.06	0.05	−1.31	0.20
Gender (male)	2.14	1.20	1.78	0.08
Education	0.15	0.57	0.26	0.79
Monthly income	−0.00	0.00	−0.76	0.45
Help-seeking Factors				
Time to help-seeking	0.05	0.03	1.97	0.06
Perceived difficulty	0.48	0.58	0.83	0.41

*R*^2^ = 0.52, F(10, 42) = 4.59, *p* < 0.001. ADCS-ADL = Alzheimer’s Disease Cooperative Study-Activities of Daily Living; NPI-Q = Neuropsychiatric Inventory-Questionnaire.

**Table 4 ijerph-18-02956-t004:** Linear regression models of carer negative emotions.

Predictors	*β*	*SE*	*T*	*p*
Person with Dementia Characteristics				
Age	0.02	0.04	0.53	0.60
Gender (male)	−0.72	0.80	−0.89	0.37
ADCS-ADL	−0.01	0.02	−0.46	0.65
NPI-Q severity	0.25	0.06	4.14	<0.00
Carer Characteristics				
Age	−0.03	0.03	−0.89	0.38
Gender (male)	−0.54	0.71	0.76	0.45
Education	0.22	0.34	0.65	0.52
Monthly income	−0.00	0.00	−1.20	0.24
Help-seeking Factors				
Time to help-seeking	0.00	0.02	0.20	0.84
Perceived difficulty	0.36	0.34	1.05	0.30

*R*^2^ = 0.46, F(10, 42) = 4.59, *p* = 0.002. ADCS-ADL = Alzheimer’s Disease Cooperative Study-Activities of Daily Living; NPI-Q = Neuropsychiatric Inventory-Questionnaire.

**Table 5 ijerph-18-02956-t005:** Ordinal regression model of perceived help-seeking difficulty.

Predictors	Estimate	SE	OR	*p*
Persons with Dementia Characteristics				
Age	0.05	0.03	1.05	0.17
Gender (male)	0.49	0.67	1.14	0.46
Carer Characteristics				
Age	−0.02	0.02	0.98	0.48
Gender (male)	0.14	0.61	1.64	0.82
Education	−0.15	0.29	0.86	0.61
Monthly income	−0.00	0.00	0.99	0.17
Perceived Barriers				
Cost concerns	2.03	0.96	7.67	0.03
Inadequate knowledge about symptoms	1.34	0.64	3.83	0.04
Difficulty in accessing service	5.66	1.37	287.20	<0.00
Relying on oneself	1.17	1.04	3.20	0.26
Lack of time to seek help	0.75	1.01	2.13	0.45

## Data Availability

No data are available. The ethical approval and participant consent for this study do not allow sharing of data beyond the research team.

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
