# Peer review of "Perceived Help-Seeking Difficulty, Barriers, Delay, and Burden in Carers of People with Suspected Dementia"

_ijerph, 2021, doi:10.3390/ijerph18062956_

Round 1
Reviewer 1 Report
This is interesting paper looking at the relationship between help-seeking difficulty and caregiver burden. The paper is well written, and I have only a few suggestions.
Methods:
It is not clear where the categories of help seeking barriers came from. There is some reference to some of the barriers in the introduction (e.g. respecting autonomy of PWD I could relate to the category ‘unwillingness to disclose the person’s problem’s; difficulty accessing services, insufficient knowledge and stigma are also referred to the introduction), but not others (e.g. concern over fees/costs, disapproval by family or friends, having to rely on oneself etc.). It would be helpful to the reader to know why the categories were chosen. This could be addressed in either the introduction or methods sections. I was also unclear about the category ‘having to rely on oneself’ and would have benefitted from a more detailed explanation of what this means (e.g. does this refer to carers not trusting formal services? Or Carers feeling they have responsibility and should rely only on themselves?). I struggled to understand the meaning of this category and how to interpret it, but this may be resolved when addressing where categories had come from.
It wasn’t clear why the NPI-Q carer distress subscale was not included in the analysis. Since this subscale seems to fit the topic of the paper, particularly in relation to carer burden, I suggest including a sentence to clarify why it was decided not to include this subscale.
There is no information on how missing data is treated in the analysis (i.e. were participants excluded if missing data, or was data imputed etc.) and this should be added.
Results:
Table 1 – Note 1 says that there is education data missing from two participants with n = 108, however the carer education data appears to be missing from 9 participants, with n = 101, rather than 108 (or 110)? This information should be added to or explained in note 1.
Section 3.2 (L150-155) – It would be helpful for the reader to have a description of how the different help-seeking difficulty levels differed for each carer burden variable, not just the statistical results.
Table 5 – typo: “Replying” instead of “Relying”
Discussion:
L199-205: My understanding of the explanation of why BSPD did not contribute to self-criticism is that carers don’t consider BPSDs to be their responsibility in the same way they do help-seeking? I think it would be helpful for the reader for this to be more explicitly stated (if correct). The explanation given doesn’t specifically refer to self-criticism which appears to be the anomalous result.
Author Response
We thank the reviewer for the time and helpful comments. Please see the attachment for our point-by-point response to your comments.
This is interesting paper looking at the relationship between help-seeking difficulty and caregiver burden. The paper is well written, and I have only a few suggestions.
Our response: Thank you very much for the very helpful comments and suggestions. We have tried our best to address them.
Methods:
It is not clear where the categories of help seeking barriers came from. There is some reference to some of the barriers in the introduction (e.g. respecting autonomy of PWD I could relate to the category ‘unwillingness to disclose the person’s problem’s; difficulty accessing services, insufficient knowledge and stigma are also referred to the introduction), but not others (e.g. concern over fees/costs, disapproval by family or friends, having to rely on oneself etc.). It would be helpful to the reader to know why the categories were chosen. This could be addressed in either the introduction or methods sections. I was also unclear about the category ‘having to rely on oneself’ and would have benefitted from a more detailed explanation of what this means (e.g. does this refer to carers not trusting formal services? Or Carers feeling they have responsibility and should rely only on themselves?). I struggled to understand the meaning of this category and how to interpret it, but this may be resolved when addressing where categories had come from.
Our response: Thank you. We have now added more detailed explanation about the basis of these categories in the Introduction and Methods. We would also want to clarify that as some of these known categories may not be most accurately captured using direct questioning about perceived barriers (e.g., sense of helplessness), some commonly ways of expression in Chinese (“having to rely on oneself”) were used to reflect these categories.
It wasn’t clear why the NPI-Q carer distress subscale was not included in the analysis. Since this subscale seems to fit the topic of the paper, particularly in relation to carer burden, I suggest including a sentence to clarify why it was decided not to include this subscale.
Our response: Thank you for the helpful suggestion. We have included the NPI-Q to reflect BPSD symptom severity; while we agree carer distress is relevant, because of the close conceptual linkage with carer burden (measured using CZBI-short), to minimise the potential overlap and circularity in our regression model, we have decided to exclude this subscale. We have now added a sentence to clarify this.
There is no information on how missing data is treated in the analysis (i.e. were participants excluded if missing data, or was data imputed etc.) and this should be added.
Our response: Complete case analysis (listwise deletion) was used with no data imputation. We have now added this information in the Methods section.
Results:
Table 1 – Note 1 says that there is education data missing from two participants with n = 108, however the carer education data appears to be missing from 9 participants, with n = 101, rather than 108 (or 110)? This information should be added to or explained in note 1.
Our response: Thank you for pointing this out. This is now added in note 1.
Section 3.2 (L150-155) – It would be helpful for the reader to have a description of how the different help-seeking difficulty levels differed for each carer burden variable, not just the statistical results.
Our response: Thank you. We have now added the descriptives.
Table 5 – typo: “Replying” instead of “Relying”
Our response: Thank you for pointing this out. We have revised this typo.
Discussion:
L199-205: My understanding of the explanation of why BSPD did not contribute to self-criticism is that carers don’t consider BPSDs to be their responsibility in the same way they do help-seeking? I think it would be helpful for the reader for this to be more explicitly stated (if correct). The explanation given doesn’t specifically refer to self-criticism which appears to be the anomalous result.
Our response: Yes, this is our interpretation too. We have now stated this more explicitly in the discussion, including the following lines: “… as shown in the regression model for self-criticism, where perceived difficulty in help-seeking significantly contributed to carer self-criticism, while BPSDs contributed to role strain and negative emotions but not self-criticism.” “On the other hand, carers may not consider BPSDs to be their responsibility in the same way they do with help-seeking.”
Reviewer 2 Report
Thank you for the opportunity to review this article. It is extremely important because the caregivers situation of people with dementia are not colorful. Due to numerous inconveniences such as social stigma or healthcare standards, they experience social exclusion, which makes their situation even more difficult. Therefore, this type of publication is extremely important. Perhaps it is thanks to them that we will be able to increase social awareness, which will entail changes in the healthcare system. Certainly, research of this type can serve as the basis for training and education in this field. As much as I wanted to, I did not have the intellectual capacity to find major inaccuracies in this study. All the important variables listed in the model (Cost concerns, Inadequate knowledge about symptoms, Difficulty in accessing service) are systemic problems that, in theory, should be dealt with in the future. However, to do this, we must pay attention to studies such as the above. Hence, I would like to ask you to publish this article after minor revision and I congratulate the authors on their interesting observations.
Minor issue
- In the title, write seeking in capital letters.
- Please standardize the subjects in the tables as you describe them once as Carer Characteristics, another time as Caregiver Characteristics. The same with PwD Characteristics and Person with Dementia Characteristics.
Author Response
We thank the reviewer for the time and helpful comments. Please see the attachment for our point-by-point response to your comments.
Reviewer 2
Thank you for the opportunity to review this article. It is extremely important because the caregivers situation of people with dementia are not colorful. Due to numerous inconveniences such as social stigma or healthcare standards, they experience social exclusion, which makes their situation even more difficult. Therefore, this type of publication is extremely important. Perhaps it is thanks to them that we will be able to increase social awareness, which will entail changes in the healthcare system. Certainly, research of this type can serve as the basis for training and education in this field. As much as I wanted to, I did not have the intellectual capacity to find major inaccuracies in this study. All the important variables listed in the model (Cost concerns, Inadequate knowledge about symptoms, Difficulty in accessing service) are systemic problems that, in theory, should be dealt with in the future. However, to do this, we must pay attention to studies such as the above. Hence, I would like to ask you to publish this article after minor revision and I congratulate the authors on their interesting observations.
Our response: We sincerely thank the reviewer for the encouraging comments.
Minor issue
- In the title, write seeking in capital letters.
Our response: Thank you. We have revised accordingly.
- Please standardize the subjects in the tables as you describe them once as Carer Characteristics, another time as Caregiver Characteristics. The same with PwD Characteristics and Person with Dementia Characteristics.
Our response: We have now checked and ensured consistency throughout.
Reviewer 3 Report
Title Perceived Help-seeking Difficulty, Barriers, Delay, and Burden 2 in Carers of People with Suspected Dementia
This manuscript examined the relationship between perceived help-seeking difficulty and carer burden, and the barriers contributing to perceived difficulty. The authors concluded that public awareness about symptoms, support in navigating service, and financial support may reduce carer burden, which is linked with perceived difficulty in help-seeking.
However, the results are difficult to understand and “Discussion” is not necessarily based on the present results.
Followings are my comments to authors.
Introduction
The results of Table 2-4 indicated which predictors such as patients’ characteristics, carer characteriscs, and help-seeking factors were significantly associated with the three factors of care burden such as self-criticism, carers’ role strain, and carers’ negative emotions. Furthermore, the results of Table 5 revealed that which predictors such as patients’ characteristics, carer characteriscs, and perceived barriers were significantly associated with help-seeking difficulty. However, it is difficult to expect the above results from the last paragraph of introduction describing the aims of this study. Please revise the aims of this study.
Methods
Please describe who made a diagnosis of dementia and the numbers of several dementia such as Alzheimer’s disease, dementia with Lewy body, vascular dementia, and front-temporal dementia. It is particularly important who made a diagnosis of dementia. Because definite diagnosis of dementia caused by neurodegeneration require neuropathological findings, clinical diagnosis of dementia should be made by neurologist or psychiatrist who is professional in clinical practice of dementia. The clinical symptoms dramatically differ between several sub-types of dementia, which significantly contribute to care burden and difficulty in help-seeking.
Results
- The relationships between the severity of perceived help-seeking difficulty and the three factors of care burden are unclear.
- The significant correlations between help-seeking factors and BPSD, ADL might raise the issue of multicollinearity in performing linear regression model, because help-seeking factor, NPI-Q severity, and ADCS-ADL are used as predictors according to Table 2-4.
- If authors examined the correlation between perceived help-seeking difficulty and ADL, BPSD, and care burden, authors should include ADL, BPSD, and care burden as a predictor in regression model of perceived help-seeking difficulty in Table 5.
- Please clearly demonstrate that Table 2-4 examined which predictors including help-seeking factors are significantly associated with care burden (self-criticism, carers’ role strain, and carers’ negative emotions), while Table 5 examined which predictors including perceived barriers are significantly associated with help-seeking difficulty.
- Overall, the relationships between patients’ characteristics (BPSD, ADL), caregiver burden, and help-seeking factors are complicated and these factors are analyzed intermingled, which make the interpretation of this manuscript very difficult.
Discussion
- Line 190-192 “, and suggested perceived help-seeking difficulty as a related stressor on top of other known stressors such as BPSDs and ADL functioning of the person with dementia.” →Which results (such as Table2-5) indicate above conclusion?
- Paragraph 3 (Line 206-223): Although the contents of this paragraph might be plausible, the description of this paragraph is not based on the present results.
- Paragraph 5 (Line 232-243): The description of this paragraph is not based on the present result. If authors would like to discuss the relationships between BPSDs and carer burden, authors should discuss based on the results of Table 2 and 3, in which BPSDs are significantly associated with carers’ role strain and negative emotions, respectively.
Author Response
We thank the reviewer for the time and helpful comments. Please see the attachment for our point-by-point response to your comments.
Title Perceived Help-seeking Difficulty, Barriers, Delay, and Burden 2 in Carers of People with Suspected Dementia
This manuscript examined the relationship between perceived help-seeking difficulty and carer burden, and the barriers contributing to perceived difficulty. The authors concluded that public awareness about symptoms, support in navigating service, and financial support may reduce carer burden, which is linked with perceived difficulty in help-seeking.
However, the results are difficult to understand and “Discussion” is not necessarily based on the present results.
Followings are my comments to authors.
Our response: We are grateful for the reviewer’s constructive comments. We hope we have been able to address these concerns.
Introduction
The results of Table 2-4 indicated which predictors such as patients’ characteristics, carer characteriscs, and help-seeking factors were significantly associated with the three factors of care burden such as self-criticism, carers’ role strain, and carers’ negative emotions. Furthermore, the results of Table 5 revealed that which predictors such as patients’ characteristics, carer characteriscs, and perceived barriers were significantly associated with help-seeking difficulty. However, it is difficult to expect the above results from the last paragraph of introduction describing the aims of this study. Please revise the aims of this study.
Our response: We have added these objectives after the aims of the study to hopefully clarify things and prepare the readers.
Methods
Please describe who made a diagnosis of dementia and the numbers of several dementia such as Alzheimer’s disease, dementia with Lewy body, vascular dementia, and front-temporal dementia. It is particularly important who made a diagnosis of dementia. Because definite diagnosis of dementia caused by neurodegeneration require neuropathological findings, clinical diagnosis of dementia should be made by neurologist or psychiatrist who is professional in clinical practice of dementia. The clinical symptoms dramatically differ between several sub-types of dementia, which significantly contribute to care burden and difficulty in help-seeking.
Our response: We agree with the reviewer about the importance of accuracy and timeliness of a dementia diagnosis and its subtyping. We would however wish to clarify that the current study focuses on an early detection service that forms part (“pre-diagnostic assessment”) of a triage system (see [18, 28]), with the carer interview happening upon reaching this service, with the clinical diagnosis confirmation being a subsequent event (“Only those whose assessment results subsequently led to a confirmed diagnosis of dementia were included”). Thus, it would be stretching our data a bit for us to comment on the impact of a definitive diagnosis has on carer burden and perceived help-seeking difficult. How the subsequent diagnostic process, including who made the diagnosis, impacts on carer burden and perceived help-seeking difficulty would be another important area for further studies (and require reference to clinical guidance regarding subspecialities qualification, competence, and role within the healthcare team that vary across countries and services settings). We have now included this point in the Discussion.
Results
- The relationships between the severity of perceived help-seeking difficulty and the three factors of care burden are unclear.
Our response: Thank you. A correlation matrix of the variables is now included in the supplementary table.
- The significant correlations between help-seeking factors and BPSD, ADL might raise the issue of multicollinearity in performing linear regression model, because help-seeking factor, NPI-Q severity, and ADCS-ADL are used as predictors according to Table 2-4.
Our response: We agree that multicollinearity might be a concern, for which we have checked and all VIF values in the models ranged between 1.07 and 1.26, suggesting no serious issue with multicollinearity. We have now included this point in the results.
- If authors examined the correlation between perceived help-seeking difficulty and ADL, BPSD, and care burden, authors should include ADL, BPSD, and care burden as a predictor in regression model of perceived help-seeking difficulty in Table 5.
Our response: We thank the reviewer for the suggestion. In the supplementary table that we have now included, perceived help-seeking difficulty is correlated with ADL, BPSD, and aspects of carer burden. The reason for not including all correlated variables in the model, however, was considering the limit in number of predictors with the sample size, and that the research question (which we have now specified in the introduction) to identify the more salient reported barriers contributing to help-seeking difficulty. We agree with the reviewer that it would be important for future studies to include larger samples sufficiently powered to allow simultaneously testing the various inter-acting variables based on the stress process model, using more advanced statistical methods such as structural equation modelling, and incorporating help-seeking factors identified in this study. We have now elaborated more about this point in the limitations.
- Please clearly demonstrate that Table 2-4 examined which predictors including help-seeking factors are significantly associated with care burden (self-criticism, carers’ role strain, and carers’ negative emotions), while Table 5 examined which predictors including perceived barriers are significantly associated with help-seeking difficulty.
Our response: We have now tried to explain the research questions relevant to the analyses more clearly.
- Overall, the relationships between patients’ characteristics (BPSD, ADL), caregiver burden, and help-seeking factors are complicated and these factors are analyzed intermingled, which make the interpretation of this manuscript very difficult.
Our response: We agree with the reviewer that it would be important for future studies to include larger samples sufficiently powered to allow simultaneously testing the various inter-acting variables based on the stress process model, using more advanced statistical methods such as structural equation modelling, and incorporating help-seeking factors identified in this study. We have now elaborated more about this point in the limitations.
Discussion
- Line 190-192 “, and suggested perceived help-seeking difficulty as a related stressor on top of other known stressors such as BPSDs and ADL functioning of the person with dementia.” →Which results (such as Table2-5) indicate above conclusion?
Our response: It is based on the results reported in Table 2. We have now elaborated the explanation in text: “help-seeking difficulty as a related stressor on top of other known stressors such as BPSDs and ADL functioning of the person with dementia, as shown in the regression model for self-criticism, where perceived difficulty in help-seeking significantly con-tributed to carer self-criticism, while BPSDs contributed to role strain and negative emotions but not self-criticism”.
- Paragraph 3 (Line 206-223): Although the contents of this paragraph might be plausible, the description of this paragraph is not based on the present results.
Our response: We have added a line to explain that this is a discussion arising from our findings: “The finding that perceived help-seeking difficulty contributed to carer burden in terms of self-criticism deserves attention to the possible nature of their relationship. In a previous study, …”
- Paragraph 5 (Line 232-243): The description of this paragraph is not based on the present result. If authors would like to discuss the relationships between BPSDs and carer burden, authors should discuss based on the results of Table 2 and 3, in which BPSDs are significantly associated with carers’ role strain and negative emotions, respectively.
Round 2
Reviewer 3 Report
This manuscript was revised correctly and might be suitable for publication in its current form.